# Stress and the Consumption of Ultra-Processed Foods during COVID-19’s Social Distancing: Are Mental Disorders Mediators in This Association? ELSA-Brasil Results

**DOI:** 10.3390/nu16132097

**Published:** 2024-06-30

**Authors:** Raphaela Kistenmacker Pires, Rosane Harter Griep, Patricia de Oliveira da Silva Scaranni, Arlinda B. Moreno, Maria del Carmen B. Molina, Vivian C. Luft, Maria de Jesus Mendes da Fonseca, Leticia de Oliveira Cardoso

**Affiliations:** 1Postgraduate Program in Public Health Epidemiology, Escola Nacional de Saúde Pública, Fundação Oswaldo Cruz, Rio de Janeiro 21040-900, Brazil; raphaelakpires@gmail.com; 2Laboratory of Education in Environment and Health, Instituto Oswaldo Cruz, Fundação Oswaldo Cruz, Rio de Janeiro 21040-900, Brazil; 3Gaffrée and Guinle University Hospital, Federal University of the State of Rio de Janeiro, Rio de Janeiro 20270-00, Brazil; patricia.oliveira.silva@hotmail.com; 4Department of Epidemiology and Quantitative Methods in Health, Escola Nacional de Saúde Pública, Fundação Oswaldo Cruz, Rio de Janeiro 21040-900, Brazil; morenoar@uol.com.br (A.B.M.); mariafonseca818@gmail.com (M.d.J.M.d.F.); leticiadeoliveiracardoso@gmail.com (L.d.O.C.); 5Postgraduate Program in Nutrition and Health, Universidade Federal do Espírito Santo, Vitoria 29075-910, Brazil; mdcarmen2007@gmail.com; 6Postgraduate Program in Epidemiology, Faculdade de Medicina, Universidade Federal do Rio Grande do Sul, Porto Alegre 90035-003, Brazil; vcluft@hcpa.edu.br

**Keywords:** COVID-19 pandemic, mental disorders, ultra-processed foods, NOVA classification, structural equation modeling (SEM)

## Abstract

The COVID-19 pandemic exacerbated various determinants of mental disorders. Several behavioral changes were observed given this increase, including harmful health consequences, such as excessive consumption of ultra-processed foods (UPFs). To assess this relationship, we investigated the meditational role of symptoms of mental disorders (anxiety and depression) in the association between stress resulting from social distancing during the COVID-19 pandemic and the consumption of UPFs in 3884 Brazilian public employees in a supplementary study of the ELSA-Brasil study. Structural equation models (SEMs) were estimated to assess the direct and indirect effects mediated by symptoms of mental disorders. The results suggested a significant and positive mediational effect of the symptoms of mental disorders on the association between the stress resulting from social distancing and the consumption of UPFs. These findings contribute to informing the need for policies and early interventions in potentially stressful situations, with a focus on the promotion of mental health, and may thus help prevent or reduce the consumption of unhealthy foods.

## 1. Introduction

According to the Global Burden of Disease study (GBD) 2019 [1], even before the COVID-19 pandemic emerged in 2020, mental disorders were among the main causes of the global burden of diseases, with approximately one billion people worldwide living with some psychiatric disorder and with depression and anxiety leading the list. The COVID-19 pandemic exacerbated this scenario, as the rapid spread of the virus prompted the World Health Organization (WHO) and numerous national governments to implement strategies to control the novel disease and mitigate its impact on populations. These strategies, aimed at delaying the epidemic’s spread and preventing the collapse of healthcare systems, included lockdowns and social distancing measures [2,3,4,5,6,7].

Despite the extreme importance of stay-at-home decisions to guarantee physical health, the adoption of social distancing guidelines had significant social and economic repercussions. For instance, the global economy experienced a severe downturn, with an estimated loss of 114 million jobs worldwide in 2020 alone [8,9]. These economic hardships, coupled with reduced social interactions and increased isolation, significantly amplified its adverse psychological effects, including heightened financial concerns, diminished social support, and increased feelings of loneliness, depression, and anxiety [10,11,12]. Reflecting these conditions, the GBD 2020 estimated a more than 25% increase in depression and anxiety disorders during the first year of the pandemic, with even higher rates in countries facing greater challenges in managing COVID-19, such as Brazil [13].

In response to the surge in depression and anxiety during the pandemic [14,15], several studies have reported that individuals adopted behavior changes, with detrimental health consequences. These include increased consumption of alcoholic beverages [16,17] and cigarettes [18], as well as negative changes in physical activity [19], sleep [19], and food consumption [20,21,22]. Specifically, regarding food consumption, although previous studies have highlighted negative changes, they have not thoroughly investigated the underlying reasons behind these, especially in relation to the increased consumption of ultra-processed foods (UPFs).

UPFs are classified within the NOVA system, which categorizes foods based on the degree and purpose of their industrial processing. These foods are typically industrial formulations made from ingredients derived from foods or synthesized from other organic sources, often lacking whole foods and being high in fat, salt, or sugar while being deficient in dietary fiber, protein, various micronutrients, and other bioactive compounds [23]. Examples of UPFs include sodas, frozen or fast food French fries, instant noodles or soup, ice cream, and chocolates. These products are designed to be highly palatable and appealing, making them a common coping mechanism for stress and emotional relief [24].

Excessive consumption of UPFs has been associated with various adverse health outcomes, including obesity, cardiovascular diseases, diabetes, and certain types of cancer [25,26,27,28,29]. Given the potential negative impact of UPFs on health, it is essential to understand the factors contributing to their increased consumption during the pandemic, particularly those related to stress and mental health. However, no study to date has investigated the relationship between the stress induced by social distancing and the consumption of UPFs.

In this context, we hypothesize that the stress resulting from social distancing during the COVID-19 pandemic is associated with excessive consumption of UPFs and that this relationship may be mediated by symptoms of mental disorders, such as anxiety and depression. Therefore, we aim to fill this knowledge gap using a structural equation modeling approach to evaluate (i) the association between stress resulting from social distancing and the consumption of ultra-processed foods and (ii) whether symptoms of mental disorders mediate the association between stress resulting from social distancing and the consumption of ultra-processed foods.

## 2. Materials and Methods

### 2.1. Study Design, Data Collection, and Study Population

The study’s data source was the Brazilian Longitudinal Study of Adult Health (ELSA-Brasil), a multicenter cohort study designed to investigate the incidence and risk factors for NCDs, mainly cardiovascular diseases and diabetes. The study population consists of public employees active at or retired from teaching and research institutions in six Brazilian state capitals (Federal University of Espírito Santo—UFES; Federal University of Minas Gerais—UFMG; Federal University of Bahia—UFBA; Federal University of Rio Grande do Sul—UFRGS; the University of São Paulo—USP; and Oswaldo Cruz Foundation—FIOCRUZ) [30,31]. Further details on the sample, recruitment, and data collection in the ELSA-Brasil study are available in previously published articles [32]. The baseline (2008–2010) included 15,105 participants of 35 to 74 years of age who underwent clinical tests and interviews. Participants returned to the study centers for the first (2012–2014; n = 14,014) and second (2017–2019; n = 12,636) follow-up visits. In addition, ELSA-Brasil has a system for follow-up and annual investigation of health events, such as hospitalizations and novel diagnoses, as well as deaths.

From June 2020 to March 2021, the participants at the second follow-up visit, except those from the University of São Paulo, totaling 8442, were invited to participate in the supplementary study (ELSA-COVID) to assess the short- and medium-term impacts of COVID-19. Of these, 5639 participants (67%) answered the online questionnaire via an app developed specifically for this study or by telephone contact, assisted by a trained professional. Our analyses only considered participants who reported that they followed the social distancing recommendations (totally or partially; n = 4593), and our study also excluded individuals with missing data for the target variables, leading to a final total of 3884 participants (Figure 1). This study was conducted according to the guidelines laid down in the Declaration of Helsinki, and all procedures involving human subjects/patients were approved by the National School of Public Health Ethics Committee (approval number 60384522.0.0000.5240)

### 2.2. Study Variables

#### 2.2.1. Exposure: Stress Resulting from Social Distancing

Stress resulting from social distancing was assessed with a continuous ordinal variable, resulting from the score on the question “On a scale from 1 to 10, how stressful has it been for you to maintain social distancing?”, with 1 for “not at all stressful” and 10 for “extremely stressful”.

#### 2.2.2. Mediator: Mental Disorders

The presence of symptoms related to depression and anxiety was assessed according to the participants’ answers to two of the three subscales comprising the Depression, Anxiety, and Stress Scale—Short Form (DASS-21), a scale already translated into and validated for Brazilian Portuguese [33]. DASS-21 consists of three Likert-scale subscales, each with four points ranging from 0 (“Does not apply to me at all”) to 3 (“Applies to me very much or most of the time”). Each subscale consists of seven items aimed at assessing symptoms of depression, anxiety, and stress in the seven days prior to the interview [33]. The subscale scores were calculated by adding the items’ scores and multiplying them by 2 to correspond to the original scale (DASS-42). The result of these scores in continuous format was used for the analyses in the structural equation model. In the bivariate analyses, the results were categorized for depression as “No” (0–9 points) versus “Yes (≥10 points) and for anxiety as “No” (0–7 points) versus “Yes” (≥8 points) [34].

#### 2.2.3. Outcome: Consumption of UPFs

To assess the consumption of UPFs, we used data from the lists of foods consumed on the previous day, with yes or no answers from the participants. We added positive responses for consumption on the previous day of foods classified according to NOVA [23] as ultra-processed, including 21 groups of foods and beverages: sodas; industrialized juice (carton, bottle, or concentrated); powdered artificial juice; yogurt with added fruit and sugar; sausages, hamburgers, or nuggets; ham, baloney, prosciutto, salami, etc.; sliced bread (white or whole grain); margarine; frozen or fast food French fries; mayonnaise, ketchup, or mustard; readymade salad dressing; instant noodles or soup; frozen or fast-food pizza; frozen lasagna or other readymade frozen meals; packaged salty snacks or potato chips; crackers; cookies (with or without filling); packaged cupcakes; cereal bars; popsicles or brand ice cream; and chocolate bars or bonbons.

#### 2.2.4. Covariables

The following variables were used: sex (female, male), schooling (master’s/PhD, undergraduate/specialization, and secondary school or less), and age (continuous and categorized as <55 years, 55–64 years, or 65 years and older).

### 2.3. Statistical Analysis

Descriptive analyses were performed with absolute and relative frequencies for the qualitative variables and means and standard deviations for the quantitative variables. T-tests were performed for comparison of the means, adopting a *p*-value < 0.05 for significant differences. Spearman’s correlation coefficients were used to investigate the associations between the continuous variables and to determine the final mediation model to be tested. 

Structural equation models (SEM) were estimated to assess the direct effect of the stress resulting from social distancing (St_dis) on the consumption of ultra-processed foods (UPFs), as well as the indirect effect, mediated by symptoms of anxiety and depression. We applied the FIML (full information maximum likelihood) estimator, which uses robust estimates for non-normality, since the variables did not display a normal distribution. We created a latent variable (symptoms of mental disorders, S_MD) explained by computing the depression and anxiety items from DASS-21. In the confirmatory factor analysis, the standardized factor loads were significant and greater than 0.70, indicating good convergent validity, thus representing an adequate latent variable.

The fits of the SEMs were assessed using the following indicators and respective reference values: chi-square with *p* > 0.05; CFI (Comparative Fit Index) > 0.95; TLI (Tucker–Lewis Index) > 0.95; SRMR (Standardized Root Mean Square Residual) < 0.05; RMSEA (Root Mean Square Error of Approximation) < 0.05; RMSEA 90%CI (confidence interval), for which the upper limit could not exceed 0.08. *p*-values < 0.05 were considered significant [35,36].

All the analyses were performed with the R software, version 4.3.2. Analysis of the SEMs used the lavaan package, version 0.6-16 [37].

## 3. Results

The final sample consisted of 3884 participants, 59% of whom were women, 37% of whom had master’s or PhD degrees, and 41% of whom were in the 55–64-year age group. The prevalence rates for symptoms of depression and anxiety were 21% and 15%, respectively. We observed higher means on the scale of stress resulting from social distancing (St_dis), with statistically significant differences, in women, in participants with less schooling, younger individuals, and in those with symptoms of depression and anxiety. For consumption of UPFs, we saw higher mean levels in men and in participants with less schooling and with symptoms of depression and anxiety (Table 1).

Table 2 shows the correlations between stress resulting from social distancing (St_dis), symptoms of depression (SympDepr) and anxiety (SympAnx), consumption of UPFs, and age. The results suggest that greater stress resulting from social distancing correlated significantly with younger age. In addition, stress resulting from social distancing was positively associated with symptoms of depression and anxiety, besides increased consumption of UPFs.

Given these results of correlations and to analyze the mediational role of symptoms of mental disorders in the stress resulting from social distancing on the consumption of UPFs, we estimated a structural equation model. The model produced goodness-of-fit indices that suggest a good fit in the data: *p*-value X2 = 0.396; CFI = 1.00; TLI = 1.001; SRMR = 0.003; RMSEA = 0.000 [90%CI = 0.000–0.04]. The mediation model parameters are presented in Figure 2.

The analysis of the standardized results of the direct, indirect, and total effects of the stress resulting from social distancing on the consumption of UPFs showed a statistically significant and positive association for an indirect effect (β = 0.024; *p*-value = 0.001), a positive association but without statistical significance for a direct effect (β = 0.019; *p*-value = 0.269), and a positive and significant association for the total effect (β = 0.043; *p*-value = 0.008). These results thus suggest a significant and positive mediational effect of symptoms of mental disorders on the association between the stress resulting from social distancing and the consumption of UPFs.

## 4. Discussion

Our results showed that the association between stress resulting from social distancing and the consumption of UPFs appears to be best explained by the symptoms of mental disorders (depression and anxiety) in the active or retired public employee participants of the ELSA-Brasil study during the pandemic.

Although this study was the first of its kind to investigate the mediational role of symptoms of mental disorders in the effect of stress resulting from social distancing during the COVID-19 pandemic on the consumption of UPFs, a study in Chile [38] and another in Portugal [39] also identified an effect of the mediation of symptoms of mental disorders (measured by DASS-21). However, the first study longitudinally examined whether these symptoms mediated the relationship of the perceived impact of COVID-19 on comfort food consumption in a convenience sample of 1048 students and university staff (academic and non-academic) [38]. And the second study explored their mediational role in the association between the experienced psychosocial impact of the COVID-19 pandemic crisis during lockdown and disordered eating behaviors in adults during the first COVID-19 lockdown period in Portugal [39]. Although they had distinct outcomes, both studies agreed that threatening situations such as the pandemic could trigger emotional suffering, leading to coping mechanisms related to food consumption. 

As a result of the stress and negative emotions experienced during the pandemic, studies also concluded that many people turned to comfort food, consuming foods not out of necessity or hunger but as a way of coping with their feelings, and mostly opting for larger amounts of food such as candies and sweets, fatty foods, and salty snacks [40].

In Brazil, an increase in the consumption of UPFs was documented [22,41,42], and one study indicated an association between the prevalence of symptoms of anxiety and depression and its impact on increased consumption of UPFs during this period [43]. Additionally, a web survey conducted in the initial period of the pandemic (24 April to 24 May 2020) found that Brazilians with a prior diagnosis of depression (self-reported) displayed higher odds of increasing their consumption of UPFs during the pandemic [20]. Despite these studies hypothesizing that the increased consumption of UPFs may also be linked to the negative impact of social distancing on the population’s health, our study is the first study to provide support for this mediational role in the Brazilian context.

Our results from the correlational analysis suggested that a greater perception of stress from social distancing was found in women, younger individuals, and those with less schooling, corroborating studies that have assessed the psychological effects of sheltering at home and social distancing due to COVID-19 [12]. These effects have mostly been related to factors such as fear of infection, boredom, financial losses, inadequate management of the pandemic by government authorities, and a lack of (or excess) information [12,44].

Importantly, the fact that the sample consisted of public employees in the ELSA-Brasil study limits any generalization of the results to the overall Brazilian population. Another limitation corresponds to the data collection period, which involved distinct moments, ranging from rigorous lockdown to a certain loosening of social distancing measures. Although research has increasingly evidenced an association between anxiety and depression and inadequate food consumption and its impact on increased consumption of UPFs, it is important to emphasize that these associations express bidirectional relations [45]. Thus, future analyses adopting a longitudinal design are necessary to elucidate the temporal sequence of symptoms of mental disorders and the consumption of UPFs. However, this study used a robust statistical method to assess the mediation and identified the direct, indirect, and total effects, thereby revealing mental disorders as key factors in the causal path analyzed here.

## 5. Conclusions

This study’s findings highlighted how stressful situations such as social distancing measures can trigger emotional suffering, leading to coping strategies involving inadequate food consumption. The results underscore the need for policies and early interventions in stressful situations analogous to the COVID-19 pandemic, with a focus on the promotion of mental health, thus potentially contributing to the prevention of or a reduction in the consumption of unhealthy foods.

## Figures and Tables

**Figure 1 nutrients-16-02097-f001:**
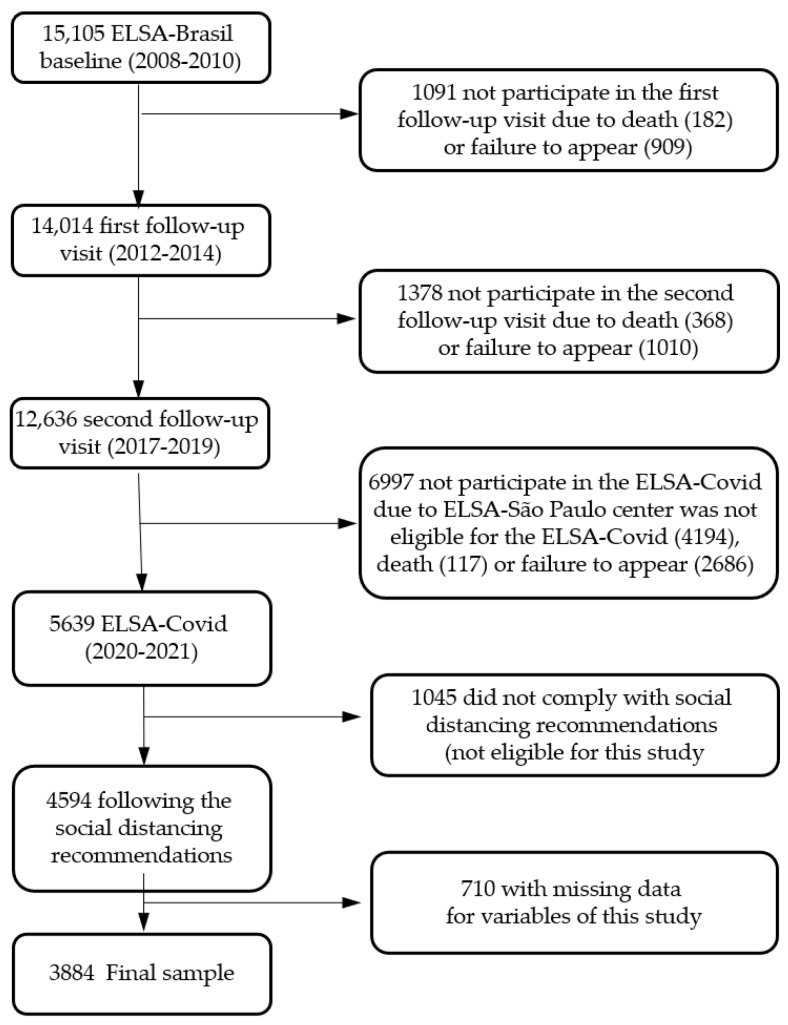
Flowchart of participants in the present study.

**Figure 2 nutrients-16-02097-f002:**
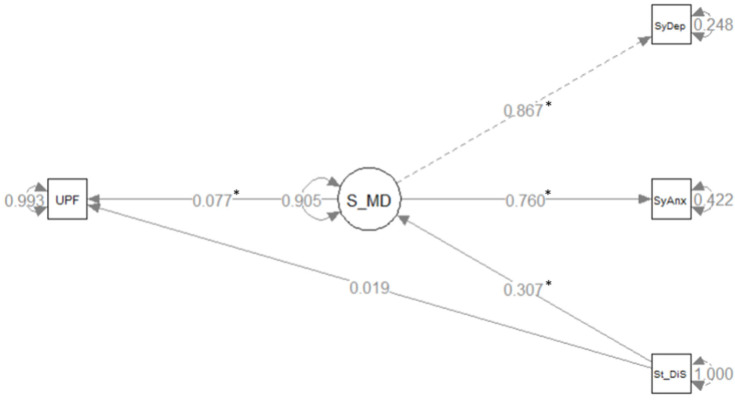
Structural equation model assessing the standardized regression coefficients of the effects of stress resulting from social distancing (St_dis) on the consumption of ultra-processed foods (UPFs) through symptoms of mental disorders (S_MD). Note: * *p*  <  0.01 ELSA-COVID (2020–2021), (N = 3884).

**Table 1 nutrients-16-02097-t001:** Characteristics of the study population according to stress resulting from social distancing (St_dis) and consumption of ultra-processed foods (UPFs). ELSA-COVID (2020–2021), (N = 3884).

			St_dis	UPFs
	n	%	Mean (sd)	Mean (sd)
**Variables**	3884		4.76 (2.73)	2.02 (1.64)
**Sex**				
Male	1591	41.0	4.56 (2.67) ***	2.21 (1.71) ***
Female	2293	59.0	4.90 (2.77)	1.90 (1.58)
**Schooling**				
≤complete secondary	1063	27.4	5.02 (2.98) ***	2.15 (1.69) *
Undergraduate/specialization	1374	35.4	4.77 (2.67)	2.01 (1.63)
Master’s/PhD	1447	37.3	4.55 (2.57)	1.95 (1.61)
**Age group (years)**				
<55	1131	29.1	5.01 (2.65) ***	2.09 (1.62) **
55–64	1588	40.9	4.81 (2.70)	1.92 (1.63)
>64	1165	30.0	4.44 (2.82)	2.11 (1.67)
**Symptoms of depression**				
No	3082	79.4	4.46 (2.67) ***	1.98 (1.62) **
Yes	802	20.6	5.91 (2.65)	2.20 (1.71)
**Symptoms of anxiety**				
No	3290	84.7	4.54 (2.67) ***	2.00 (1.63) *
Yes	594	15.3	5.96 (2.76)	2.18 (1.67)

* *p* < 0.05; ** *p* < 0.01; *** *p* < 0.001.

**Table 2 nutrients-16-02097-t002:** Means, standard deviations (SDs), and correlations between stress resulting from social distancing (St_dis), symptoms of depression (SympDepr) and anxiety (SympAnx), consumption of UPFs, and age. ELSA-COVID (2020–2021), (N = 3884).

	Descriptives		Spearman’s Correlation Coefficients			
	Mean	SD	St_dis	SympDepr	SympAnx	UPFs
St_dis	4.76	2.73	1 ***			
SympDepr	5.07	6.59	0.28 ***	1 ***		
SympAnx	3.27	4.74	0.23 ***	0.54 ***	1 ***	
UPF	2.02	1.64	0.05 **	0.05 ***	0.08 ***	1 ***
Age	59.91	8.63	−0.09 ***	−0.1 ***	−0.03 *	0.01

Note: Spearman’s correlation coefficients were used to calculate the association between variables. * *p* < 0.05; ** *p* < 0.01; *** *p* < 0.001.

## Data Availability

The data presented in this study are available on request from the corresponding author. The data are not publicly available due to the commitment to maintain confidentiality and secrecy with the database made available by the ELSA-Brasil Project, as well as all information related to the project entitled.

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
