# Peer review of "Stress and the Consumption of Ultra-Processed Foods during COVID-19’s Social Distancing: Are Mental Disorders Mediators in This Association? ELSA-Brasil Results"

_nutrients, 2024, doi:10.3390/nu16132097_

Round 1

Reviewer 1 Report

Comments and Suggestions for Authors

I hope this letter finds you well. I am writing to provide feedback on the recent publication of The mediational role of mental disorders in the association between stress resulting from social distancing and the consumption of ultra-processed foods: results from the ELSA-Brasil Study in Nutrients. As a conscientious reader, I have taken the liberty to offer constructive criticism to enhance the scholarly quality and impact of the work.

1. The article would greatly benefit from a more elaborate exposition of its research purpose and a clearer articulation of specific research questions. While the introduction provides a foundational overview, expanding upon this section with more detailed objectives would enhance reader comprehension and engagement.

2. If the article aims to utilize a survey research methodology within the social sciences, it would be advantageous to incorporate a more expansive literature review and conduct a deeper analysis of the existing body of knowledge. Specifically, delving deeper into the literature surrounding mental disorders, the societal impacts of COVID-19, and the discourse on the consumption of processed food products would enrich the study's theoretical framework. Additionally, a more robust research design, including a systematic approach to hypothesis formulation, would strengthen the study's methodological integrity.

3. The absence of clearly formulated research hypotheses is notable as it limits the study's ability to adequately address the influence of potential mediating variables. This deficiency raises concerns about the reliability and validity of the study's findings, underscoring the importance of establishing well-defined hypotheses to guide the research process.

4. The section dedicated to research findings and subsequent discussion could benefit from a more in-depth analysis and a more extensive exploration of connections with existing literature. By fostering a deeper dialogue with previous research, the study could offer valuable insights and contribute meaningfully to the academic discourse in its respective field.

5. Notably absent from the study are discussions regarding its limitations and potential practical implications. Addressing the limitations of the research methodology and acknowledging the constraints of the study would provide transparency and help readers interpret the findings more accurately. Furthermore, identifying potential practical implications and avenues for future research would enhance the study's relevance and scholarly impact, making it a more valuable contribution to the academic community.

Thank you for considering my feedback. I trust that these suggestions will be received in the spirit of academic collaboration and improvement. 

Author Response

Comments 1: The article would greatly benefit from a more elaborate exposition of its research purpose and a clearer articulation of specific research questions. While the introduction provides a foundational overview, expanding upon this section with more detailed objectives would enhance reader comprehension and engagement.

Response 1: Thank you for pointing this out. We agree with this comment and have revised the introduction of the manuscript accordingly. These changes are highlighted and can be found on the first page, from lines 62 to 83.

 Comments 2: If the article aims to utilize a survey research methodology within the social sciences, it would be advantageous to incorporate a more expansive literature review and conduct a deeper analysis of the existing body of knowledge. Specifically, delving deeper into the literature surrounding mental disorders, the societal impacts of COVID-19, and the discourse on the consumption of processed food products would enrich the study's theoretical framework. Additionally, a more robust research design, including a systematic approach to hypothesis formulation, would strengthen the study's methodological integrity.

 Response 2: We would like to clarify that our study utilized an epidemiological approach. While structural equation modeling (SEM) has traditionally been used more in research in the social sciences and humanities, in recent years it has become a robust technique widely utilized and consolidated in epidemiological studies. However, we have acknowledged the need to make our hypotheses clearer, and we have highlighted them in the introduction, on page 1, line 75.

 Comments 3: The absence of clearly formulated research hypotheses is notable as it limits the study's ability to adequately address the influence of potential mediating variables. This deficiency raises concerns about the reliability and validity of the study's findings, underscoring the importance of establishing well-defined hypotheses to guide the research process.

 Response 3: Thank you for pointing this out. We agree with this comment. As mentioned above, we have clarified our study hypothesis, highlighting it in the introduction on page 1, line 75.

Comments 4: The section dedicated to research findings and subsequent discussion could benefit from a more in-depth analysis and a more extensive exploration of connections with existing literature. By fostering a deeper dialogue with previous research, the study could offer valuable insights and contribute meaningfully to the academic discourse in its respective field.

Response 4: Agree. We have, accordingly, modified our discussion to emphasize this point.

Comments 5: Notably absent from the study are discussions regarding its limitations and potential practical implications. Addressing the limitations of the research methodology and acknowledging the constraints of the study would provide transparency and help readers interpret the findings more accurately. Furthermore, identifying potential practical implications and avenues for future research would enhance the study's relevance and scholarly impact, making it a more valuable contribution to the academic community.

 Response 5: Although we had already mentioned our limitations in the discussion, we made some changes to emphasize the potential practical implications. These changes are highlighted on page 6, line 259.

Reviewer 2 Report

Comments and Suggestions for Authors

The proposed article proposes the results of a study the association between stress resulting from social distancing and the consumption of ultra-processed foods, in Covid 19 pandemic.

This is an old study but other related articles have already been published so what is the real main purpose of this?

I believe these results do not bring something innovate.

Somer questions should be answered, such as:

What is the main question addressed by the research? only social distance and UPF? nothing else more consistent?

Do you consider the topic original or relevant in the field? 

What does it add to the subject area compared with other immense interesting published?

I believe the title is too big for this article

I believe this study should be completed with other interesting aspects.

Author Response

Response: Thank you for your feedback. We have acknowledged the need to make our hypotheses clearer, and, accordingly, modified our introduction to emphasize this point. As highlighted, this study was the first of its kind to investigate the mediational role of symptoms of mental disorders in the effect of stress resulting from social distancing during the COVID-19 pandemic on consumption of UPF. We hope this clarifies the significance of our research in contributing to the existing literature.

We firmly believe that the topic of our study is original and highly relevant in the field. We investigated exposure in an innovative way, such as the perception of stress due to social distancing, as well as the mediating role that symptoms of mental disorders had in the relationship with the consumption of UPF. We consider this research of great relevance since its high consumption is associated with the development of chronic noncommunicable diseases (NCDs), including cardiovascular diseases, diabetes, and certain types of cancer.

We agree that the title is long, but we intended to highlight what is innovative about the article. Anyway, we made a change.

After addressing the questions you raised, we hope our manuscript will encompass a broader range of perspectives.

Reviewer 3 Report

Comments and Suggestions for Authors

Interesting study on anxiety and depression and UPF. First of all the authors should report the no. of ethic committee acceptance. They should refer to sample, recruitment and data collection since previous reference is dated back in 2013.

Why publish after 3 years? Study has been conducted between 2020 and 2021.

Only 3 variables were used. Why didn't the authors use more variables?

Please calculate cronbach a for all variables used.

Results are not critically discussed and are poorly given in 2 tables. These results should be enormous.

Introduction needs to be strengthened.

Comments on the Quality of English Language

Minor editing

Author Response

First of all the authors should report the nº. of ethic committee acceptance.

Thank you for your feedback. We acknowledge the importance of reporting the number of ethics committee acceptance in our manuscript. We would like to clarify that we have already included the ethics committee approval number at the end of the article under the 'Institutional Review Board Statement.' As per your request, we will also incorporate this information into the main body of the text. These changes have been made and can be found on page 3, from lines 112 to 115.

They should refer to sample, recruitment and data collection since previous reference is dated back in 2013.

Our study utilized data from a larger cohort study that has been following 15,105 participants since 2008. The sample selection was conducted at the inception of the study. To enhance clarity, we have included a flowchart on page 3 depicting the selection process.

Why publish after 3 years? Study has been conducted between 2020 and 2021.

The delay in publication is due to the nature of our study, which is derived from a larger ongoing research initiative. Data availability for analysis can be time-consuming in such comprehensive studies. Additionally, this article forms part of the doctoral thesis of the first author, and as you might be aware, doctoral programs, particularly in Brazil where they typically span around 4 years, require substantial time for completion.

Furthermore, we identified a valuable opportunity to contribute to a special issue ('Eating Habits and Nutritional Aspects during the COVID-19 Pandemic') dedicated to this thematic focus in a prestigious journal. This special issue aligns closely with our research objectives, prompting us to submit our findings at this juncture.

Only 3 variables were used. Why didn't the authors use more variables?

In addition to the primary variables of interest, we also considered several sociodemographic variables that could act as confounders in the association under investigation. This approach is consistent with previous studies in the field (see for example https://doi.org/10.3390/nu13061910 and , https://doi.org/10.1007/s40519-021-01128-1 ).

The selection of variables was guided by both theoretical considerations and the need to control for potential confounding factors. By focusing on a subset of variables, we aimed to provide a clear and focused analysis of the specific relationships of interest.

Please calculate cronbach a for all variables used.

We appreciate your attention to the reliability of our measures.

In the context of Structural Equation Model (SEM) analyses, confirmatory factor analysis (CFA) plays a critical role in validating latent constructs, such as symptoms of mental disorders in our case (SympDepr and SympAnx converging into Symptoms of Mental Disorders - S_MD).

CFA is specifically designed to confirm or refute pre-existing hypotheses about how observed variables relate to latent constructs. Unlike exploratory factor analysis (EFA), which identifies latent constructs based solely on statistical patterns, CFA requires a priori expectations of which variables should load onto specific constructs. This methodological approach ensures that the measurement model accurately reflects the underlying theoretical framework.

As detailed on page 04, line 168 of our manuscript, we performed confirmatory factor analysis and found that standardized factor loadings exceeded 0.70, indicating strong convergent validity and supporting the formation of a coherent latent variable.

While Cronbach's alpha is commonly used for assessing internal consistency in scales with multiple items, its effectiveness diminishes when applied to constructs with a small number of observed variables, as in our study where we utilized only two observed variables (SympDepr and SympAnx). In such cases, SEM provides a more robust framework for evaluating model adequacy and overall fit.

Based on the substantial factor loadings, significance tests, and overall model fit indicators presented in Figure 2 on page 06, line 219, we conclude that our measurement model demonstrates adequate fit and validity. This substantiates our theoretical framework that symptoms of depression and anxiety converge into a single latent construct representing symptoms of mental disorders.

Results are not critically discussed and are poorly given in 2 tables. These results should be enormous.

Thank you for your feedback regarding the presentation and discussion of results in our manuscript. We acknowledge that the larger study from which our data is derived yields extensive results. However, for the purpose of this article, we focused on pertinent findings that directly contribute to achieving our specific objectives.

Given the innovative nature of our study, our discussion centered on these key findings and their implications. We aimed to provide a focused analysis that highlights the novel aspects and contributions of our research within the scope of the study's objectives.

While the full scope of results from the larger study is vast, our approach ensured that the results presented in the two tables are essential for understanding the outcomes relevant to our research questions. These tables were designed to succinctly convey the core findings necessary to support our conclusions and contribute to the field.

Introduction needs to be strengthened.

We agree that the introduction needed to be strengthened, and we have made the necessary revisions to enhance its clarity and depth. The updated introduction now provides a more comprehensive context and highlights the significance and originality of our study.

We have highlighted the changes in the revised manuscript for your convenience.

Thank you for your constructive comments, which have helped us improve the quality of our work

Round 2

Reviewer 1 Report

Comments and Suggestions for Authors

No more comments.

Author Response

No more comments.

Reviewer 2 Report

Comments and Suggestions for Authors

After these improvements I believe that this modified version may be accepted for publication.

Author Response

Thank you for your positive feedback on the modifications we made based on your suggestions. We're glad to hear that you believe the revised version is now suitable for publication. We appreciate your time and valuable input throughout this process.

Reviewer 3 Report

Comments and Suggestions for Authors

AUTHORS HAVE REVISED SUFFICIENTLY AND PAPER CAN BE ACCEPTED AS IT IS

Comments on the Quality of English Language

Minor editing